# Formal and informal care received by middle-aged and older adults with chronic conditions in Canada: CLSA data

**Wei Zhang**[1,2]*, **Huiying Sun**[2]

**1** School of Population and Public Health, University of British Columbia, Vancouver, British Columbia, Canada, **2** Centre for Health Evaluation and Outcome Sciences, Vancouver, British Columbia, Canada

* wzhang@cheos.ubc.ca

**Data Availability Statement:** The CLSA data used in our study is third-party data. The CLSA data are available from the Canadian Longitudinal Study on Aging (www.clsa-elcv.ca) for researchers who meet the criteria for access to de-identified CLSA

## Abstract

### Background

Middle-aged and older adults are more likely to suffer from chronic conditions, which can increase their need for both formal and informal care. This study seeks to assess and compare the extent to which the use of formal and informal care is attributed to different chronic conditions among middle-aged and older women and men in Canada.

### Methods

We used baseline data from the Canadian Longitudinal Study on Aging (CLSA). Outcomes of interest were the number of hours of formal care and informal care received during the past 12 months. All chronic conditions were first classified according to existing classification frameworks. If total formal and informal care hours for a particular condition differed greatly from other conditions, we considered it as a stand-alone classification. We used a two-part model consisting of a logistic regression for the probability of receiving formal/informal care and a generalized linear model for the hours of formal/informal care for those who received care.

### Results

Our final analytic sample was 23,206 women and 22,903 men who did not have missing data. Among the 16 chronic conditions considered, multiple sclerosis, memory problems, Parkinsonism, and stroke had the greatest average marginal effects on overall hours of formal care among women (53.07, 13.95, 9.13 and 8.14 incremental hours annually, respectively) and men (152.17, 8.13, 13.95 and 6.00 incremental hours). Similarly, the average marginal effects of these four conditions on informal care were the greatest (77.78, 29.52, 26.18 and 34.95 incremental hours for women and 133.94, 34.99, 104.86 and 17.85 incremental hours for men).

data. We had no special access privileges to the data and other researchers will be able to access the data in the same manner as the authors. The specific data application requirements and process can be found at https://www.clsa-elcv.ca/data-access/data-access-application-process. Please email access@clsa-elcv.ca to request a Magnolia user account and other information on CLSA data application.

**Funding:** This study is supported by the Start Up Funds from the Faculty of Medicine, University of British Columbia (WZ is the principal investigator https://mednet.med.ubc.ca/Research/FundingOpportunities/Pages/New-Faculty-Research-Award.aspx). WZ is supported by a Michael Smith Foundation for Health Research Scholar Award (https://www.msfhr.org/health-work-and-society-improving-health-economic-evaluations). The funders had no role in study design, data collection and analysis, decision to publish, or preparation of the manuscript.

**Competing interests:** The authors have declared that no competing interests exist.

## Conclusions

Chronic conditions, especially multiple sclerosis, Parkinsonism, memory problems, and stroke, are associated with substantial time of formal and informal care in middle-aged and older women and men. Findings will help decision-makers assess the potential impact of chronic disease prevention and management programs in an aging population.

## Introduction

The prevalence of people with health problems, especially chronic conditions, increases with age [1]. In 2018, middle-aged (45 to 64 years) and older adults (≥65 years) made up 27.4% and 17.1% of the Canadian population, respectively [2]. Health problems not only lead to higher utilization of health care services such as physician visits, emergency department visits, and hospitalization but also increase the need of at-home care [3]. In 2018, approximately 10% (~3 millions) of Canadians aged 15 years and older received some form of at-home care, and 39% of care receivers were 65 years or older [4].

An aging population with chronic conditions could increase the need for both formal and informal care, where formal care refers to the assistance (including personal care, medical care, managing care, housework, transportation, and meal preparation) received at home from professionals or paid workers and informal care refers to the assistance received from family, friends, or neighbours [5]. In Canada, about 44% of care receivers received informal care only, 12% relied on formal care only, and the remaining received both formal and informal care [6]. Both formal and informal care contribute significantly to the healthcare system as this enables care receivers with health conditions, disabilities, or age-related needs to remain at home, thus reducing the need for other health care service utilization such as nursing home, hospital care and physician visits [7–9]. Although informal care could reduce the direct expenditure from the health sector perspective, the indirect costs incurred by informal caregivers can also be large from a societal perspective. It is important for policy makers to understand the demand for formal and informal care by different chronic conditions as this can help inform the allocation of resources to meet the increasing demands of an aging population.

Many studies have focused on estimating the time of formal/informal care received by people with one specific chronic condition including dementia [10], multiple sclerosis [11], cancer [12,13], depression [14], cardiovascular disease, and diabetes [15]. Due to differing settings and population characteristics, however, the time estimates reported by published studies vary widely and may not be comparable across different chronic conditions. Moreover, few studies have compared the use of formal and informal care across different diseases. They either simply rank the diseases by the frequencies reported as the main reason requiring formal/informal care [6,16,17], or attribute the total reported formal/informal care time to a disease if it was identified as the main reason for care by care recipients [3]. Older adults are often more likely to have multimorbidities and it may be difficult to attribute the formal or informal care received to each specific disease.

Gender differences and inequalities have been observed in previous studies. In addition to gender-specific health conditions, women are more likely to suffer from age-related health conditions due to their longer life expectancies [1]. The provision of home care especially informal care is widely feminized [18]. However, fewer studies have explored gender differences in terms of care receiving and the findings are mixed depending on study population [4,6,19,20]. Our study seeks to assess and compare the extent to which the use of formal and

informal care is attributed to different chronic conditions among middle-aged and older women and men in Canada.

## Materials and methods

### Study population

Our study population consists of all respondents who participated in the Canadian Longitudinal Study on Aging (CLSA) at baseline. The CLSA is one of the largest and most comprehensive research platforms examining health and aging [21–23]. Baseline CLSA includes 51,338 participants who were community-living women and men, aged 45 to 85 years, able to read and speak English or French, and living in one of the ten Canadian provinces at time of recruitment [21–23]. The CLSA has two categories of participants: "Tracking" participants who were followed by telephone interview only and were randomly selected within an age/sex strata from all ten Canadian provinces (n = 21,241), and "Comprehensive" participants who were followed via in-home interviews, onsite physical examinations, and biological specimen collection (n = 30,097). Comprehensive participants were randomly selected within an age/sex strata from within 25 to 50 km of 11 sites across seven Canadian provinces [21–23]. Baseline participants were recruited in 2011 to 2015 via four sources: Canadian Community Health Survey–Healthy Aging (for CLSA Tracking only), Provincial Health Registries, Telephone Sampling-Random Digit Dialing, and Quebec Longitudinal Study on Nutrition and Aging (for CLSA Comprehensive only). Exclusion criteria included persons living on First Nations reserves or settlements, full-time members of the Canadian Armed Forces, institutionalized individuals, and cognitively impaired persons unable to provide informed consent at the time of recruitment. Baseline CLSA data was collected between September 2011 and May 2015 and included information on demographics, lifestyle, behaviour, and social, physical, psychological, and health status. Our study was approved by the University of British Columbia-Providence Health Care Research Ethics Board (Ethics Certificate No. H18-01588) and was a secondary use of the CLSA data.

### Measures

We considered two outcome variables: total number of informal care hours received during the past 12 months and total number of formal care hours received during the past 12 months. We derived the first outcome by multiplying the numerical responses to the following two questions: 1) "During the past 12 months, about how many weeks did this person/these people provide you with such assistance? Include assistance from all family members, friends and neighbours in your estimate."; and 2) "How many hours per week, on average, did this person/these people provide you with assistance?" [5]. The number of hours spent on formal care was similarly calculated by multiplying the number of weeks the main professional person or organization helped the respondent by average hours per week. According to the CLSA questionnaire, the assistance for both informal care and formal care included "personal care (such as assistance with eating, dressing, bathing, or toileting), medical care (such as help taking medicine or help with nursing care, for example, dressing change or foot care), managing care (such as making appointments), help with activities (such as housework, home maintenance, or outdoor work), transportation (such as trips to the doctors or for shopping), and meal preparation or delivery" [5].

The key explanatory variables consisted of the different chronic conditions. In the CLSA, chronic conditions refer to "long-term conditions [that] are expected to last, or have already lasted 6 months or more and that have been diagnosed by a health professional". The CLSA includes a comprehensive list of chronic conditions (S1 Table). We first grouped these

conditions according to classification frameworks used by Griffith et al. [24], the CLSA questionnaires [5], and the CLSA DataPreview Portal based on ICD-10 [25]. Secondly, we selected the chronic conditions with a large number of total number of hours of formal and informal care (> 120 hours) and considered them as a stand-along classification (S2 Table). Thirdly, we compared the total number of hours of formal and informal care received by each chronic condition within each classification. If the total number of hours for a particular condition differed greatly (over ± 30 hours) from the mean total number of hours among all conditions within a classification, we excluded that condition from the classification and considered it as a stand-alone classification. We finally ensured the exclusions were consistent among men and women (S3 Table) and checked whether for each final classification, the number of women or men who received formal care and informal care were greater than 5, respectively. Each final classification was coded as a dummy variable if a study participant reported having any one chronic condition within a specific classification.

Although CLSA asked a question about sex (female versus male), we explained it as gender (women versus men). The study population was stratified by gender to incorporate the gender differences. We considered other characteristics associated with formal and informal care received and the presence of chronic conditions as potential confounders. Confounders consisted of: (i) demographic characteristics including age as a categorical variable (45–54, 55–64, 65–74, and 75+ years), and marital status (currently married or common law versus windowed, divorced, separated, or never married); (ii) socioeconomic status including household income as a categorical variable (<$20,000, $20,000-$49,999, $50,000-$99,999, $100,000-$149,999, ≥$150,000, Don't know/No answer/Refused), number of people living in a household (excluding the participant), own dwelling (yes versus no), and highest education attainment (less than secondary school graduation, secondary school graduation, some post-secondary, post-secondary graduation); (iii) health and lifestyle factors including body mass index classification [underweight/normal weight (<25 kg/m2), overweight (25–30 kg/m2), obese (>30 kg/m2)], type of smoker (never smoked, former smoker, and current smoker), and type of drinker in the past 12 months [not a drinker (did not drink in the last 12 months), occasional drinker, regular drinker (at least once a month)].

## Statistical analysis

The study sample included all subjects who provided valid responses to the questions related to the main outcomes, chronic conditions, and all potential confounders. Since the number of formal/informal care hours had a highly skewed distribution with a lot of zeros, we used a two-part model consisting of a logistic regression for the probability of receiving formal/informal care (part one) and a generalized linear model for the hours of formal/informal care among those who received formal/informal care in the past year (part two). We constructed the regression models by including dummy variables representing each of the chronic condition classifications simultaneously as well as all the potential confounders. Gamma or log-normal distributions for part two were determined using the Akaike information criterion (AIC). The parameters of the two parts were estimated simultaneously using the maximum likelihood method.

As suggested by CLSA [23], we used pooled (trimmed) inflation weights and geographic strata variables for descriptive statistics, and used analytic weights while including age, gender, and the interaction of provinces and the indication of two categories of CLSA participants (tracking and comprehensive) in all regression models. We conducted the analyses among all study population and then stratified by gender. We performed all analyses using SAS version 9.4 (SAS Institute, Cary, NC, USA).

## Average marginal effects

The overall expected hours of formal/informal care received was the product of the estimated probability of receiving formal/informal care and the estimated mean of formal/informal care hours among those who received formal/informal care based on the model parameters, respectively. For example, the average marginal effect of a particular chronic condition on the overall number of informal care hours or the average incremental informal care hours attributable to the particular chronic condition was derived by comparing the overall expected hours of informal care between two assumed populations (one assuming that all the study samples had the chronic condition and the other assuming that all the study samples did not have the chronic condition) while keeping all other conditions and characteristics unchanged. Detailed calculation steps are presented in S1 File.

# Results

## Chronic condition classification

The 36 chronic conditions included in the CLSA were first classified into 10 classifications (S1 Table). The total number of hours of formal and informal care received across the chronic conditions within each of the 10 classifications was then compared. Hypertension, stroke, multiple sclerosis, Parkinsonism, memory problems, dementia, and bowel incontinence were considered separate classifications because their total hours of formal/informal care received were either very large (> 120 hours) or different from that of other conditions in their respective classification categories (S2 Table). However, there were only 5 women with dementia who received formal care and the sample size was too small for us to include it as a dummy variable in regression analyses. Since people with dementia received a higher number of hours of formal and informal care, they should not be combined with either other conditions or the reference group. Thus, we decided to exclude them from our study population. The final number of chronic condition classifications was 16 for both women and men (S3 Table).

## Study sample and characteristics

Respondents were excluded if there were missing responses to hours of formal/informal care received (n = 395) and final chronic condition classifications (n = 2,835). The respondents with dementia (n = 87) and missing information on all potential confounders (n = 1,912) were further excluded. The final analytic sample of 46,109 respondents (23,206 women and 22,903 men) represented 12,113,952 Canadians aged 45 to 85 years. The average age of women was 60.2 years old and that of men was 59.9 years old. In terms of care received, 2.1% of women (1.7% of men) received formal care only, 10.5% (7.0%) received informal care only, and 3.0% (1.9%) received both formal and informal care in the past 12 months (Table 1 and S4 Table). The difference between women and men were statistically significant (p<0.001, S4 Table). Compared with men, women received larger number of formal care hours (7.13 hours [standard error (SE) = 1.77] for women versus 4.30 hours (SE = 0.74) for men; p = 0.142) and informal care hours (31.35 hours (SE = 2.43) versus 17.34 hours (SE = 3.51); p = 0.009) in the past 12 months. Approximately 85.6% (88.4% or 82.7%) of the study population (women or men) had at least one chronic condition. The prevalence of chronic condition classifications was different between women and men but their three most prevalent chronic condition classifications were the same (musculoskeletal system diseases, hypertension, and endocrine/metabolic system diseases).

**Table 1. Characteristics of study population.**

| Characteristics | All | Women | Men |
|---|---|---|---|
| | Mean (SE[a]) or Number (%) | Mean (SE) or Number (%) | Mean (SE) or Number (%) |
| Age (years) | 60.09 (0.07) | 60.24 (0.10) | 59.92 (0.10) |
| 45–54 | 12404 (38.3%) | 6367 (37.7%) | 6037 (39.0%) |
| 55–64 | 14984 (31.2%) | 7673 (31.1%) | 7311 (31.3%) |
| 65–74 | 10645 (18.8%) | 5269 (18.8%) | 5376 (18.9%) |
| 75+ | 8076 (11.6%) | 3897 (12.5%) | 4179 (10.8%) |
| Care received | | | |
| No home care received | 39935 (86.9%) | 19507 (84.4%) | 20428 (89.5%) |
| Formal home care only | 1032 (1.9%) | 587 (2.1%) | 445 (1.7%) |
| Informal home care only | 4046 (8.8%) | 2427 (10.5%) | 1619 (7.0%) |
| Both formal and informal home care | 1096 (2.4%) | 685 (3.0%) | 411 (1.9%) |
| Number of formal care hours among the entire population | 5.74 (0.97) | 7.13 (1.77) | 4.30 (0.74) |
| Number of formal care hours among those who received care | 132.03 (21.84) | 139.78 (34.02) | 120.63 (20.10) |
| Number of informal care hours among the entire population | 24.44 (2.68) | 31.35 (4.02) | 17.34 (3.51) |
| Number of informal care hours among those who received care | 218.75 (23.30) | 232.95 (28.98) | 196.49 (38.91) |
| Chronic condition classification | | | |
| Bowel incontinence | 896 (1.8%) | 586 (2.3%) | 310 (1.3%) |
| Cancer | 7020 (13.1%) | 3612 (14.1%) | 3408 (11.9%) |
| Cardiac | 8751 (16.8%) | 3724 (14.9%) | 5027 (18.7%) |
| Endocrine/Metabolic | 12895 (24.7%) | 7373 (28.7%) | 5522 (20.6%) |
| Gastrointestinal | 6940 (13.7%) | 4071 (15.8%) | 2869 (11.6%) |
| Genitourinary | 4790 (9.5%) | 3117 (12.6%) | 1673 (6.3%) |
| Hypertension | 16993 (33.6%) | 8099 (32.4%) | 8894 (35.0%) |
| Memory problems | 734 (1.7%) | 353 (1.6%) | 381 (1.7%) |
| Mental | 8932 (19.2%) | 5580 (23.5%) | 3352 (14.8%) |
| Multiple sclerosis | 278 (0.6%) | 200 (0.8%) | 78 (0.3%) |
| Musculoskeletal | 24503 (50.4%) | 13719 (55.6%) | 10784 (45.1%) |
| Neurological | 6368 (14.7%) | 4603 (20.9%) | 1765 (8.3%) |
| Ophthalmologic | 13781 (23.1%) | 7514 (26.4%) | 6267 (19.8%) |
| Parkinsonism | 171 (0.3%) | 58 (0.2%) | 113 (0.5%) |
| Respiratory | 7436 (15.1%) | 4221 (16.7%) | 3215 (13.4%) |
| Stroke | 743 (1.5%) | 299 (1.1%) | 444 (1.8%) |

[a]SE indicates standard error.

Reported mean, SE, and % were estimated using the pooled (trimmed) inflation weights and the geographic strata variables.

## Chronic conditions and formal/informal care hours

The unadjusted relationships between different chronic conditions and formal/informal care are presented in S5 Table. Adjusting for all study sample characteristics, all chronic conditions except hypertension and neurological diseases were associated with a statistically significant (p<0.05) increase in the odds of receiving formal care and all conditions except neurological diseases were associated with an increase in the odds of receiving informal care among women (Figs 1 and 2). A log-normal distribution for the generalized linear model was used to estimate the relationship between chronic conditions and the hours of formal/informal care among those who received care as it was preferred over a Gamma distribution based on AIC. Among women who received formal care, memory problems, multiple sclerosis, musculoskeletal diseases, and stroke were significantly associated with an increase in the hours of formal care

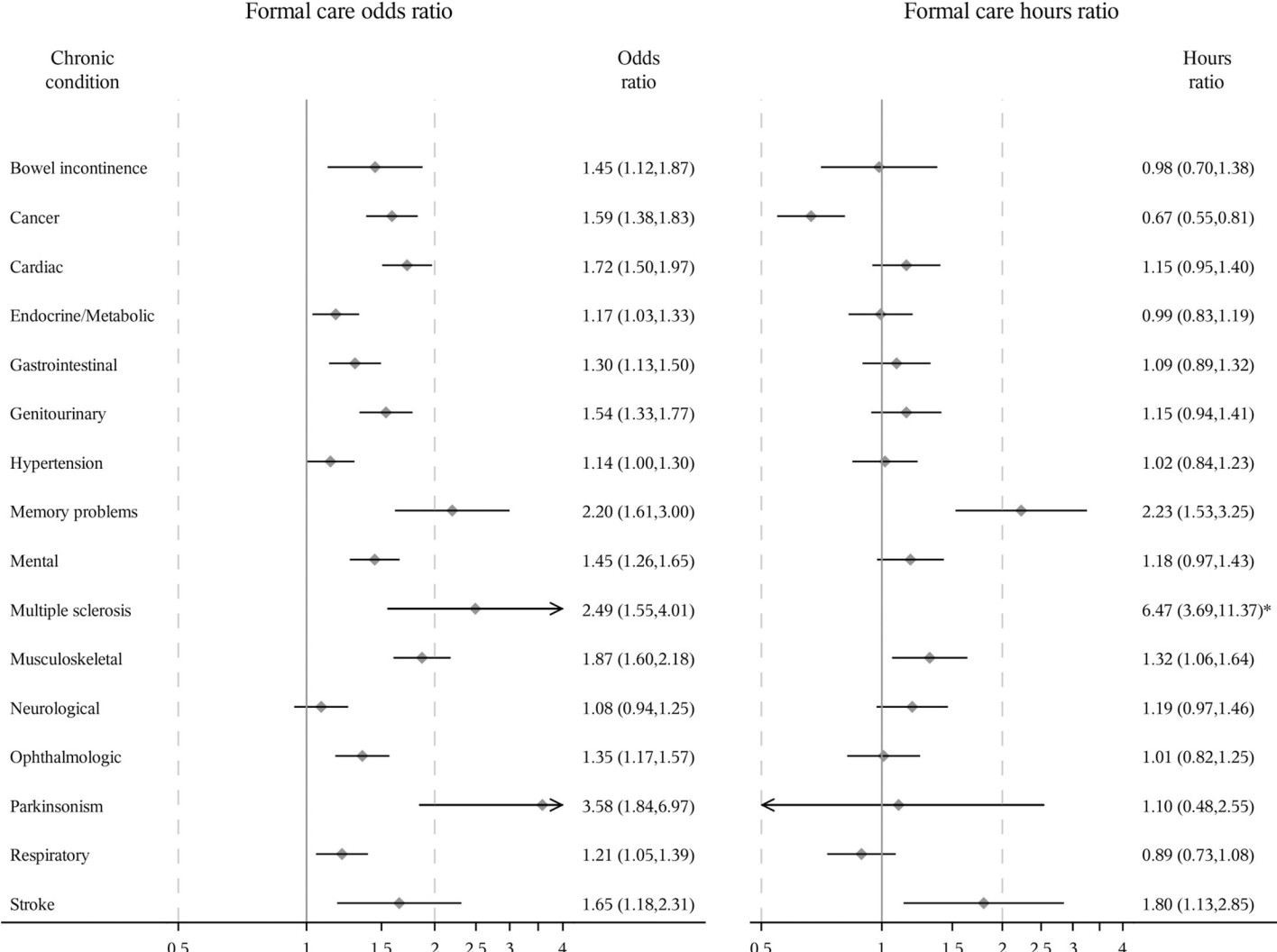

**Fig 1. Odds ratio of receiving formal care and ratio of expected hours among women receiving care.** Odds ratios [95% confidence interval (CI)] estimated using logistic regression. Ratio of expected hours (95% CI) estimated using generalized linear model with a log-normal distribution among those who received formal care, respectively. All study sample characteristics were adjusted. *Odds ratio and 95% CI not shown because they are out of range.

received but cancer was associated with a decrease in the hours of formal care received. Cancer, memory problems, mental diseases, multiple sclerosis, musculoskeletal diseases, neurological diseases and stroke were significantly associated with an increase in the hours of informal care among women who received informal care.

Slightly differently, all chronic conditions except gastrointestinal diseases and neurological diseases were associated with a statistically significant increase in the odds of receiving formal care and all conditions except bowel incontinence and neurological diseases were associated with an increase in the odds of receiving informal care among men (Figs 3 and 4). Among men who received formal care, memory problems, multiple sclerosis, neurological diseases, and stroke were significantly associated with an increase in the hours of formal care received. Cardiac diseases, genitourinary diseases, hypertension, memory problems, mental diseases, multiple sclerosis, Parkinsonism, respiratory diseases and stroke were significantly associated with an increase in the hours of informal care among men who received informal care.

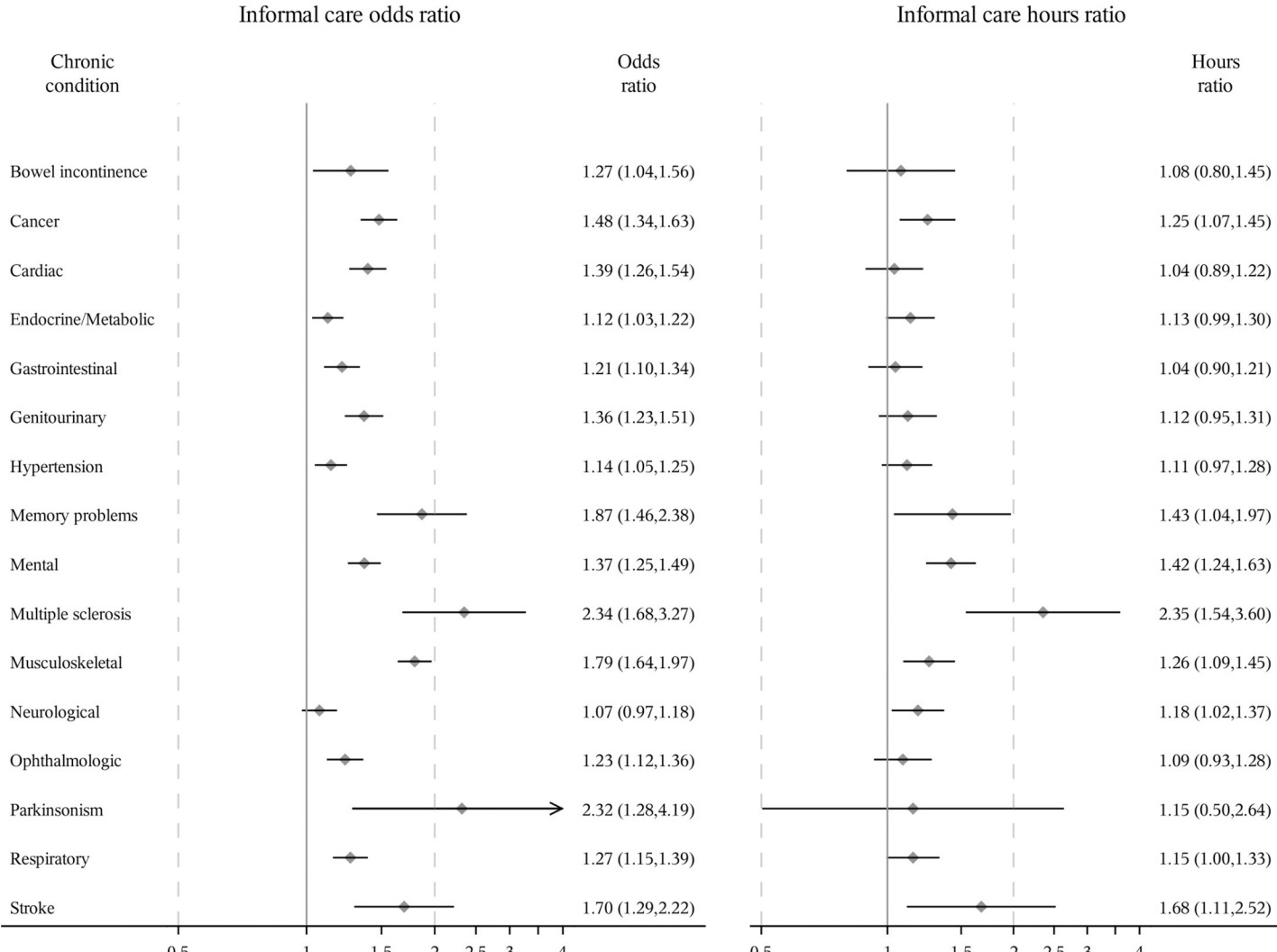

**Fig 2. Odds ratio of receiving informal care and ratio of expected hours among women receiving care.** Odds ratios [95% confidence interval (CI)] estimated using logistic regression. Ratio of expected hours (95% CI) estimated using generalized linear model with a log-normal distribution among those who received informal care, respectively. All study sample characteristics were adjusted.

Multiple sclerosis, Parkinsonism and memory problems were the three conditions with the greatest associations with the odds of receiving formal/informal care among women and men, and the hours of informal care among men who received informal care. Multiple sclerosis, memory problems and stroke were the three conditions with the greatest association with the hours of formal/informal care among women receiving formal/informal care and the hours of formal care among men receiving formal care. For example, after controlling for all other conditions and covariates, the odds of receiving formal or informal care for women with multiple sclerosis were 1.49 or 1.34 times more than those for women without multiple sclerosis, respectively. Among men, the numbers were larger (3.90 or 6.55, respectively). Among those receiving formal/informal care, women with multiple sclerosis received 6.47 or 2.35 times as many hours of formal/informal care as women without multiple sclerosis after adjusting for all other conditions and covariates. Among men, the corresponding numbers were also larger (18.86 or 3.71 times). Overall, after adjusting for all chronic conditions and other covariates, women

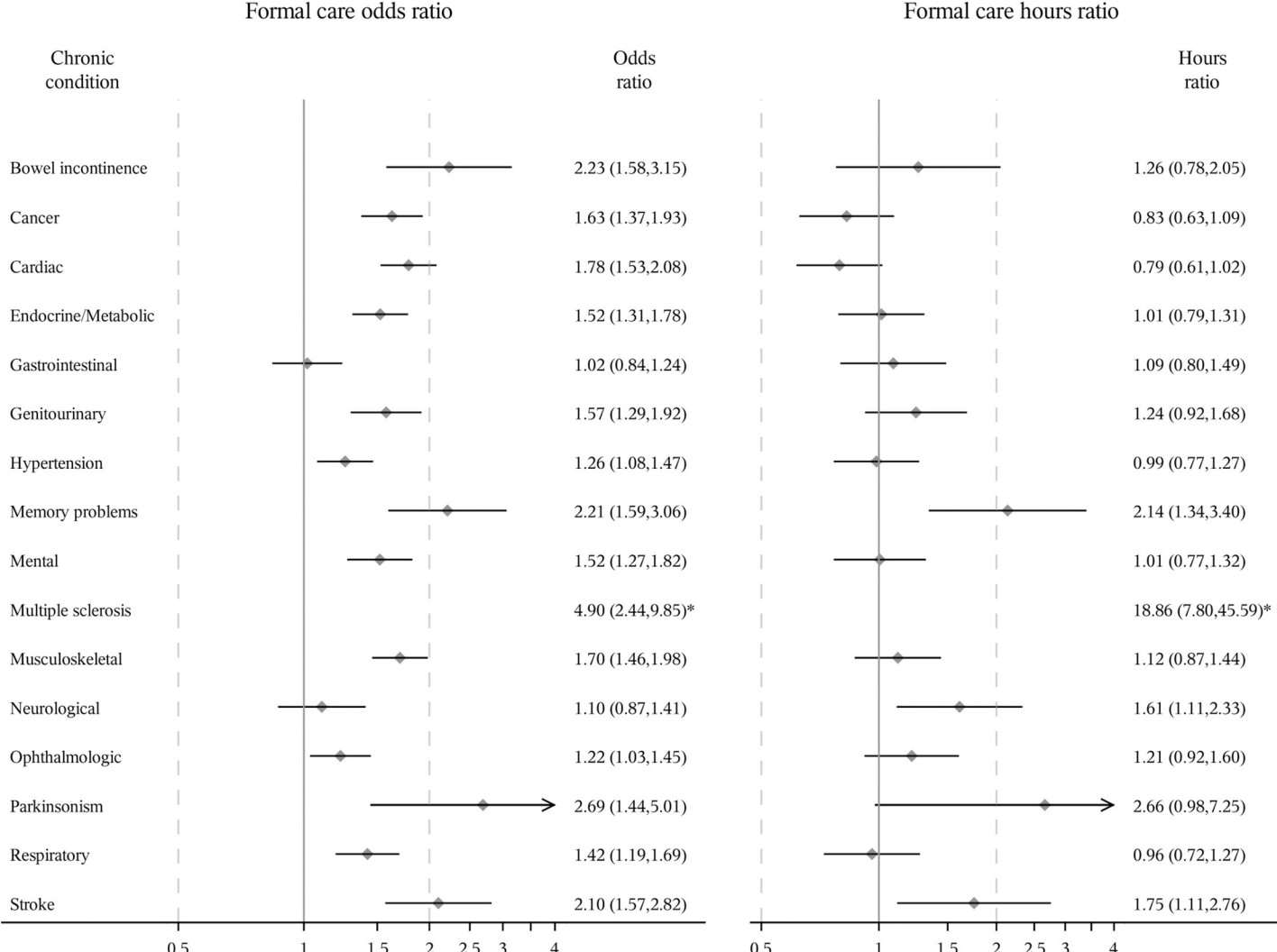

**Fig 3. Odds ratio of receiving formal care and ratio of expected hours among men receiving care.** Odds ratios [95% confidence interval (CI)] estimated using logistic regression. Ratio of expected hours (95% CI) estimated using generalized linear model with a log-normal distribution among those who received formal care, respectively. All study sample characteristics were adjusted. *Odds ratio and 95% CI not shown because they are out of range.

were more likely to receive formal care (odds ratio = 1.11 (95% confidence interval: 1.00, 1.23)) and informal care (odds ratio = 1.33 (1.25, 1.42)), and received more formal care hours (hours ratio = 1.20 (1.03, 1.41)) and informal care hours (hours ratio = 1.41 (1.27, 1.57)) among those who received care than men (S6 Table).

## Average marginal effects

Consistently, multiple sclerosis, memory problems, Parkinsonism, and stroke had the greatest average marginal effects on overall hours of formal care among women (53.07, 13.95, 9.13 and 8.14 incremental hours annually, respectively) as well as among men (152.17, 8.13, 13.95 and 6.00 incremental hours annually) (Fig 5 and S7 Table). Similarly, the average marginal effects of these four conditions on informal care were the greatest (77.78, 29.52, 26.18 and 34.95 incremental hours for women and 133.94, 34.99, 104.86 and 17.85 incremental hours for men annually, respectively). However, the magnitudes of average marginal effects were different between

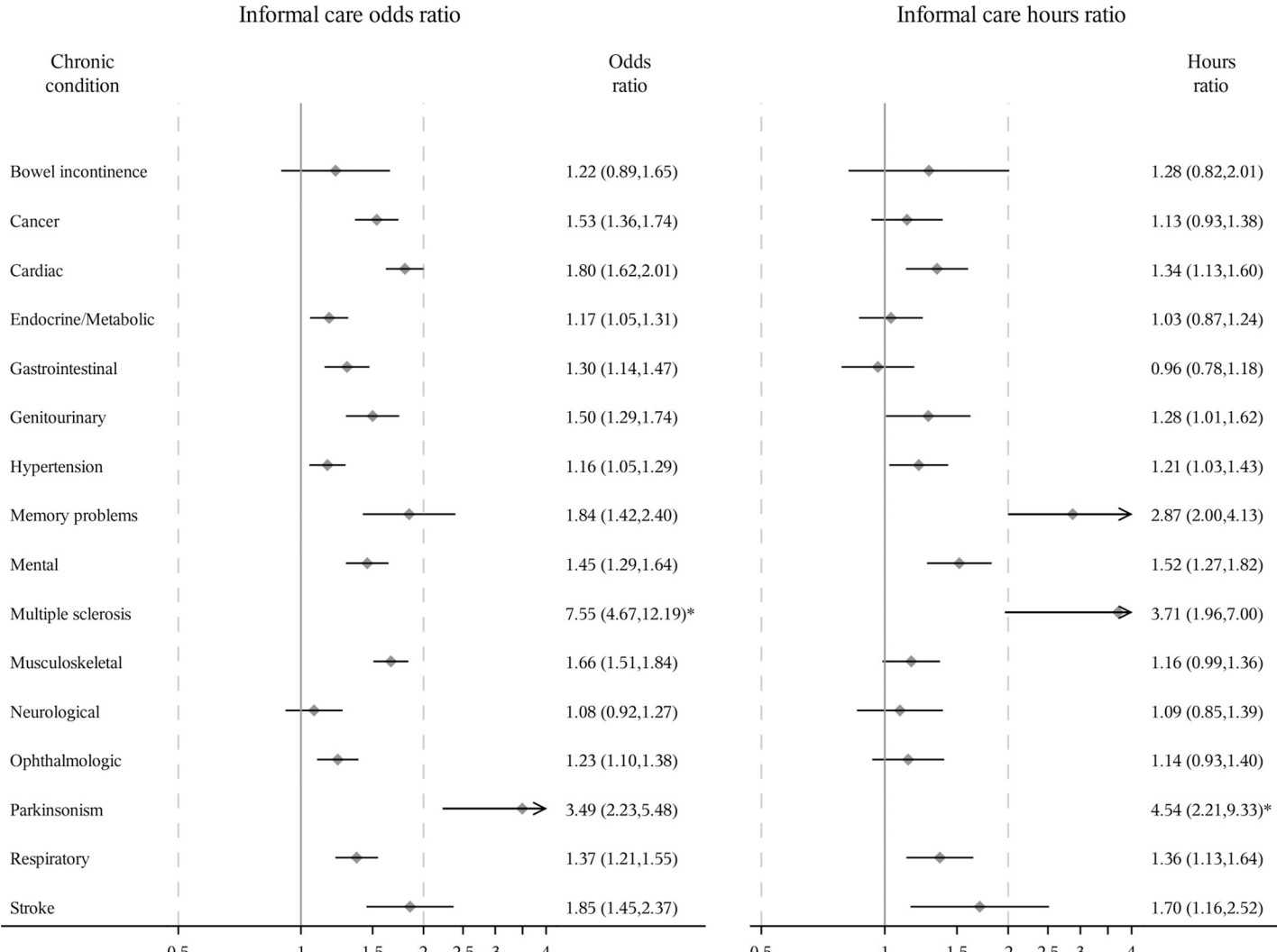

**Fig 4. Odds ratio of receiving informal care and ratio of expected hours among men receiving care.** Odds ratios [95% confidence interval (CI)] estimated using logistic regression. Ratio of expected hours (95% CI) estimated using generalized linear model with a log-normal distribution among those who received informal care, respectively. All study sample characteristics were adjusted. *Odds ratio and 95% CI not shown because they are out of range.

women and men, especially the average marginal effect of multiple sclerosis on both formal and informal care and the average marginal effect of Parkinsonism and stroke on informal care.

## Discussion

We first categorized chronic conditions by disease systems and hours of formal/informal care received and then measured the incremental effect of each chronic condition classification on formal and informal care separately, in a Canadian middle-aged and older population stratified by gender. Results show that multiple sclerosis, Parkinsonism, memory problems, and stroke were the chronic conditions requiring the most formal and informal care among both women and men. Although women overall received more formal and informal care than men, the incremental impact magnitudes of certain chronic conditions (e.g., multiple sclerosis and Parkinsonism) on formal and informal care received by men could be larger than women.

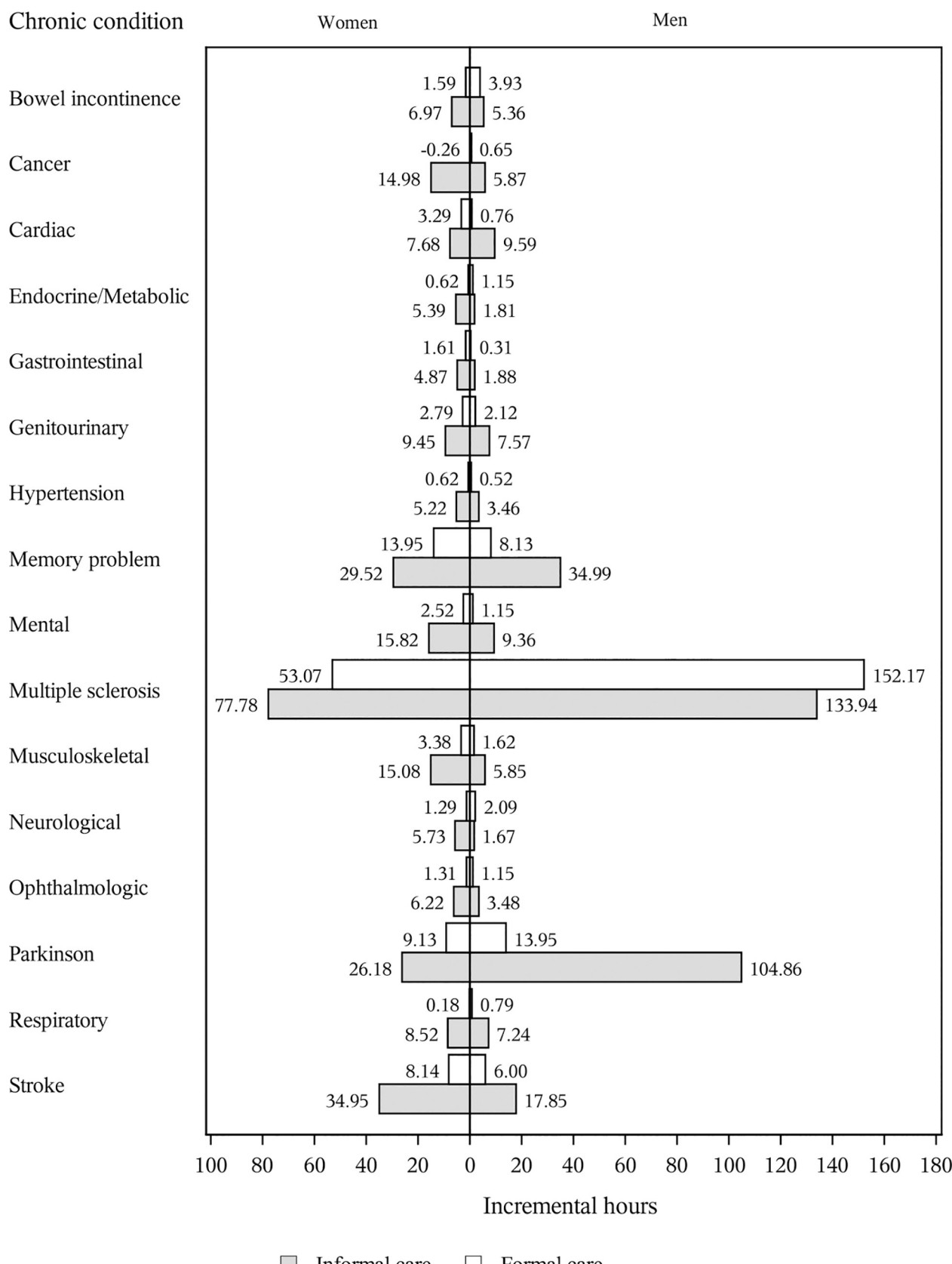

**Fig 5. Incremental number of hours of formal care and informal care.** Average incremental number of hours for each condition was estimated by taking the difference between overall expected hours (assuming all study samples had this specific condition) and overall expected hours (assuming all study samples did not have this specific condition), keeping all other conditions and characteristics unchanged.

Very few population-based studies have measured and compared the need for formal and informal care by different chronic conditions. According to a report by Statistics Canada, mental illness, accident-related injuries, aging needs, cardiovascular disease, arthritis, and cancer were the most common reasons for at-home care among Canadians aged 15 and older, and Alzheimer's disease or dementia received the most hours of care [6]. According to a report by the National Alliance for Caregiving and the AARP Public Policy Institute [16], the main presenting problems or illnesses among informal care recipients 50 years and older in the US were aging, Alzheimer's disease or dementia, mobility issues, surgery and wounds, cancer, and heart disease. The top conditions or illnesses requiring higher caregiver hours were aging, Alzheimer's disease or dementia, and cancer [16]. In our study, we also found that Alzheimer's disease or dementia (excluded due to small sample size) and cancer required a large number of informal care hours although they were not the conditions that needed the most care. However, the two reports directly asked care recipients and informal caregivers to report main problems or illnesses that required care for a limited number of conditions and illnesses. Our study improves upon this by comparing more chronic conditions and the association between such conditions and the actual hours of formal and informal care received by care recipients.

In addition, a few previous studies have investigated gender differences in the receipt of formal and informal care and the findings are not consistent. Katz et al. found that women (aged 70 years or older) with disability received more hours of formal care per week but fewer hours of informal care than men in the United States [19]. Jang and Kawachi showed that older women with disability were more likely to receive formal or informal care than men but among those who received care, men received higher average days of care per month than women in Korea [20]. In Canada, women aged 45 years and older were more like to receive care at home than men in 2012 [6] and 2018 [4]. Our study results were consistent with previous findings in Canada and showed that women received more formal care hours and informal care hours than men.

Our study has several limitations. First, the CLSA survey excluded cognitively impaired adults and thus the time of formal and informal care received is underestimated. This also resulted in very few survey participants with dementia who reported receiving formal and informal care and stopped us from including dementia as one condition in our final analyses. Our study population only represents the middle-aged and older adults without cognitive impairment and dementia in Canada. Second, the time of formal care received only captures the help from a professional person or organization that dedicated the most time and resources to helping the respondent. The time of informal care, on the other hand, captured the help from all people who provided assistance to the respondent. Thus, the hours of formal care may be underestimated and the hours of formal and informal care received are not comparable. However, this does not limit the ability to compare the need for formal care across different chronic conditions. Third, another limitation is the non-specific nature of some of the chronic conditions captured in the CLSA, resulting in a lack of sufficient information required for formal/informal care estimates. For example, CLSA does not distinguish between specific types of cancer. For such a heterogeneous group of diseases, our results may not be generalizable to the formal/informal care needed by specific cancer types. Fourth, there might be memory bias when asking about the care time (in terms of the number of weeks and the average number of hours per week) received during the previous 12 months, in which the care needs may have

been highly variable. Lastly, it is worth noticing that our study reflected the care time received by the care receivers, which could be different from the time spent on caregiving from the caregivers' perspective.

Our study has important policy implications, as it is essential for policy makers to recognize the demand for formal and informal care by gender in an aging population. Our study findings can help to prioritize resource allocation for chronic conditions by gender while addressing the increasing demand for formal and informal care in aging populations. However, our study findings could not provide clear suggestions on resource allocation between formal home care and informal care, which partially depends on whether their relationship is substitution (funding one type of care to replace the other) or complementary (funding both). The findings from previous studies on the relationship between informal care and formal home care are not consistent [26,27]. In our study, the four conditions, multiple sclerosis, memory problems, Parkinsonism, and stroke, have consistently been associated with more formal care and more informal care among both women and men, which might suggest a complementary relationship between formal and informal care. A future study is required to further investigate the relationship and whether the relationship varies by chronic conditions or functional impairment level. In addition, our study provides data that can be used to evaluate the potential cost savings when implementing chronic disease management programs from a societal perspective and can project future demand for formal and informal care.

## Conclusions

In conclusion, chronic conditions, especially multiple sclerosis, Parkinsonism, memory problems, and stroke, are associated with substantial time of formal and informal care in middle-aged and older women and men. Study findings will help decision-makers assess the potential impact of chronic disease prevention and management programs in an aging population as well as their societal value.

## Supporting information

**S1 Table. Chronic condition classification based on different methods (step one).** [a]CLSA indicates Canadian Longitudinal Study on Aging.
(DOCX)

**S2 Table. Chronic condition classification by comparing the total number of hours of formal and informal care among chronic conditions.** Bolded text indicates a chronic condition that was considered as a stand-alone classification.
(DOCX)

**S3 Table. Final chronic condition classification.** Bolded text indicates a chronic condition that was considered as a stand-alone classification.
(DOCX)

**S4 Table. Full characteristics of study population.** [a]SE indicates standard error. [b]Independent samples *t* test for continuous variables; Chi-squared test for categorical variables. Reported mean, SE, and % were estimated using the pooled (trimmed) inflation weights and the geographic strata variables.
(DOCX)

**S5 Table. Formal care and informal care by chronic conditions.**
(DOCX)

**S6 Table. Odds ratio of receiving formal/informal care and ratio of expected hours among all study population who received formal/informal care.** *p value < 0.05.
(DOCX)

**S7 Table. Incremental number of hours of formal care and informal care by chronic conditions.** [a]Average overall expected number of hours assuming all study samples were with a specific condition while keeping all other conditions and characteristics unchanged; [b]Average overall expected number of hours assuming all study samples without a specific condition while keeping all other conditions and characteristics unchanged; [c]The difference between a and b is the average incremental hours for the specific condition.
(DOCX)

**S1 File. Detailed calculation steps for average marginal effects.**
(DOCX)

## Acknowledgments

This research was made possible with data/biospecimens collected by the Canadian Longitudinal Study on Aging (CLSA). Funding for the CLSA is provided by the Government of Canada through the Canadian Institutes of Health Research under grant reference: LSA 94473 and the Canada Foundation for Innovation. This research has been conducted using the CLSA data set, Baseline Tracking, Version 3.4, and Baseline Comprehensive, Version 4.0, under Application No. 180604. The CLSA is led by Drs. Parminder Raina, Christina Wolfson, and Susan Kirkland. This study was approved by the Hamilton Integrated Research Ethics Board (Ethics Certificate No. 0164). The opinions expressed in this manuscript are the authors' own and do not reflect the views of the CLSA. We also acknowledge editorial assistance from Julie Sou.

## Author Contributions

**Conceptualization:** Wei Zhang, Huiying Sun.

**Data curation:** Wei Zhang.

**Formal analysis:** Huiying Sun.

**Funding acquisition:** Wei Zhang.

**Investigation:** Wei Zhang, Huiying Sun.

**Methodology:** Wei Zhang, Huiying Sun.

**Project administration:** Wei Zhang.

**Supervision:** Wei Zhang.

**Visualization:** Huiying Sun.

**Writing – original draft:** Wei Zhang.

**Writing – review & editing:** Huiying Sun.

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
