## [Decision Letter · Decision Letter 0]

1 Apr 2020

PONE-D-20-03193

Chronic conditions and formal and informal care among middle-aged and older adults in Canada: CLSA data

PLOS ONE

Dear Dr. Zhang,

Thank you for submitting your manuscript to PLOS ONE. After careful consideration, we feel that it has merit but does not fully meet PLOS ONE’s publication criteria as it currently stands. Therefore, we invite you to submit a revised version of the manuscript that addresses the points raised during the review process.

First, I would like to thank you for your patience while this manuscript was under review. As you will see, we now have 2 reviews of your manuscript. Overall, both are positive and both reviewers remarked on the potential contribution of this work. You will see, too, though, that both have made some suggestions for clarifications in the manuscript, in particular around the classifications of the chronic conditions and more enhanced discussion of some of the limitations (especially related to the study exclusions on dementia/cognitive impairment). You will also see that one of the reviewers has recommended that you incorporate a gender lens into the work given the overall gendered nature of care provision and differences between women and men in chronic condition profiles. This will require some re-analysis but I would strongly encourage you to consider this and at the very least, to provide more discussion of these issues throughout the manuscript.

We would appreciate receiving your revised manuscript by May 16 2020 11:59PM. To enhance the reproducibility of your results, we recommend that if applicable you deposit your laboratory protocols in protocols.io, where a protocol can be assigned its own identifier (DOI) such that it can be cited independently in the future. For instructions see: http://journals.plos.org/plosone/s/submission-guidelines#loc-laboratory-protocols

We look forward to receiving your revised manuscript.

Kind regards,

Andrea Gruneir

Academic Editor

PLOS ONE

Journal Requirements:

3. Your ethics statement must appear in the Methods section of your manuscript. If your ethics statement is written in any section besides the Methods, please move it to the Methods section and delete it from any other section. Please also ensure that your ethics statement is included in your manuscript, as the ethics section of your online submission will not be published alongside your manuscript.

Additional Editor Comments (if provided):

Reviewers' comments:

Reviewer's Responses to Questions

**Comments to the Author**

1. Is the manuscript technically sound, and do the data support the conclusions?

Reviewer #1: Yes

Reviewer #2: Yes

2. Has the statistical analysis been performed appropriately and rigorously? 

Reviewer #1: Yes

Reviewer #2: I Don't Know

3. Have the authors made all data underlying the findings in their manuscript fully available?

Reviewer #1: Yes

Reviewer #2: No

4. Is the manuscript presented in an intelligible fashion and written in standard English?

Reviewer #1: Yes

Reviewer #2: Yes

5. Review Comments to the Author

Reviewer #1: Dear authors,

your manuscript is a significant contribution to the field of formal and informal at-home care. I do have only minor comments and questions:

1. classification of chronic conditions: with reference to table S2 you describe that you exclude a particular chronic conditions from other conditions of the same class, if the estimated hours (of formal and informal) care differed greatly. This logic is hardly comprehensible. In case of your classification system in step one "circulatory conditions" you pick high blood pressure / hypertension and re-classify it finally as hypertension. Is it due to the fact that hypertension differs from the other conditions belonging to the same class, because of the low mean hours of at-home care? If this is the case, then you should have picked migraine headaches from the class of "neurological conditions" as well. You identify stroke/ CVA as a stand-alone conditions and exclude it from "circulatory conditions". The counterpart of this condition is osteoporosis within the class of "musculoskeletal conditions". Why did you not chose this conditions?

2. statistical analysis: you applied a two-part model, which I consider a great idea. Part 2 of the model was a GLM either with a gamma or a log-normal distribution. These decisions depended on the vuong test, AIC, and BIC. Why did your decisions depend on all three criterion? The vuong test is based on the BIC. The AIC might prefer a higher amount of parameters in case of large samples as in your study. Thus, it is in my eyes unnecessary to rely on all of them, especially in cases they contradict easch other.

3. incremental costs: you apply the same wages for informal care and formal/ professional care. Although you discuss this tricky point in your limitation section, I have to comment on it. You assume that you ensure the consistency between formal care and informal care using a replacement cost approach. This slightly contradicts your statement from the paragraph above, where you state that the hours of formal care may be underestimated and as a consequence the hours of formal and informal care are not comparable. I must admit that indirect costs (in this case productivity losses from a social perspective) are difficult to calculate, because it depends on the point of view. I request the authors to comment on the following two proposals: a.) calculate the indirect costs of informal care using a second approach and refer to it as supplementary material; b.) leave out the calculation of incremental costs, because it does not reinforce the meaning of the manuscript towards its policy implications.

Reviewer #2: Thank you very much for the opportunity to review this interesting manuscript. This is an original and relatively new investigation on the time (and the derived cost) of formal and informal care received by people with various chronic conditions. The manuscript is well written and the language is clear and understandable. The methodology is appropriate and well executed, the results are well expressed and the conclusions are derived from them.

However, the manuscript requires some improvements that I detail below.

- The title is not clear regarding the content of the study. Authors should specify that it is the informal and formal care received (and not provided) for adults with chronic conditions. As worded, it is confusing.

- In the summary (as in the results section of the main text) it should be clearly specified over what period of time the hours of care received (and the corresponding cost) were calculated (annual? Weekly?). Likewise, better detail in the methods how the chronic conditions suffered have been classified.

- In the introduction: reference 4 on data on the population receiving informal and / or formal care in Canada is from 2014, are there no more current data?

- On the other hand, the importance of the study is justified on the grounds that home care reduces the need for other more expensive residential care options. This argument does not take into account the indirect costs incurred by informal caregivers who help patients at home, and responds to a model of caregiver as a resource that is widely discussed.

- The manuscript does not incorporate a gender analysis, which is fundamental in the subject of home care, and informal care in particular, widely feminized. Likewise, suffering from chronic conditions and disability is more frequent in women, who also have a longer life expectancy. The manuscript should incorporate the differences and inequalities between women and men, from the introduction to the method of analysis (sex is considered as a simple adjustment variable, when it should be stratified by sex), the presentation of the data (the tables should to be presented stratified by sex), and in their discussion.

- The methodology should specify which are the categories of formal services considered in the questionnaire, and which the categories for informal care are.

- The authors would have to explain how they have analyzed the possible coexistence of different chronic conditions classified into different groups.

- I consider it important, as I said before, that the data be analyzed and presented by sex. Likewise, in Table 1 the formal and informal care time received should be presented by age group.

- As possible limitations, the following should also be discussed:

- Possible limitation of excluding cognitively impaired persons from the study population (care time is underestimated)

- Possible memory bias when asking about the care time received during the previous 12 months, in which the care needs may have been highly variable.

- Possible differences with other studies that ask about the time dedicated to caregivers, instead of people who need care. The perception of time spent can be very different.

- The authors should incorporate into the discussion some implications for the models of substitution or complementarity between formal and informal care.

- Likewise, the implications for gender equality of the study results.

6. PLOS authors have the option to publish the peer review history of their article (what does this mean?). If published, this will include your full peer review and any attached files.

Reviewer #1: Yes: Andreas Hoell

Reviewer #2: No

---

## [Author Response · Author response to Decision Letter 0]

7 May 2020

Please see our responses to reviewer and editor comments in the attached document.

---

## [Decision Letter · Decision Letter 1]

23 Jun 2020

Formal and informal care received by middle-aged and older adults with chronic conditions in Canada: CLSA data

PONE-D-20-03193R1

Dear Dr. Zhang,

We’re pleased to inform you that your manuscript has been judged scientifically suitable for publication and will be formally accepted for publication once it meets all outstanding technical requirements.

Kind regards,

Andrea Gruneir

Academic Editor

PLOS ONE

Additional Editor Comments (optional):

Thank you for taking the time to address the reviewers' comments so thoroughly and for your patience with the review process. Please be sure to do a thorough proofread of your manuscript and that you follow all of the CLSA's publication criteria.

Reviewers' comments:

Reviewer's Responses to Questions

**Comments to the Author**

1. If the authors have adequately addressed your comments raised in a previous round of review and you feel that this manuscript is now acceptable for publication, you may indicate that here to bypass the “Comments to the Author” section, enter your conflict of interest statement in the “Confidential to Editor” section, and submit your "Accept" recommendation.

Reviewer #1: All comments have been addressed

2. Is the manuscript technically sound, and do the data support the conclusions?

Reviewer #1: Yes

3. Has the statistical analysis been performed appropriately and rigorously? 

Reviewer #1: Yes

4. Have the authors made all data underlying the findings in their manuscript fully available?

Reviewer #1: Yes

5. Is the manuscript presented in an intelligible fashion and written in standard English?

Reviewer #1: Yes

6. Review Comments to the Author

Reviewer #1: The auhors revised their manuscript in a fashion that they come out with something new.

They meet all the comments of the reviewers and/ or give detailed explainations.

7. PLOS authors have the option to publish the peer review history of their article (what does this mean?). If published, this will include your full peer review and any attached files.

Reviewer #1: No

---

## [Editor Report · Acceptance letter]

25 Jun 2020

PONE-D-20-03193R1 

Formal and informal care received by middle-aged and older adults with chronic conditions in Canada: CLSA data 

Dear Dr. Zhang:

I'm pleased to inform you that your manuscript has been deemed suitable for publication in PLOS ONE. Congratulations! Your manuscript is now with our production department. 

Kind regards, 

on behalf of

Dr. Andrea Gruneir 

Academic Editor

PLOS ONE